# Inhibition of HPSE/SDC-2 axis-induced epithelial-mesenchymal transition for treating IC/BPS

**Zhengsen Chen**[1,2☯], **Yuting He**[1,2☯], **Junjie Zhang**[1,2☯], **Qingyu Ge**[1,2], **Tianpeng Du**[1,2], **Zongyao Fan**[1,2], **Junyi Zhou**[1,2], **Xin Yang**[1,2*], **Baixin Shen**[1,2*], **Zhongqing Wei**[1,2*]

1 Department of Urology, The Second Affiliated Hospital of Nanjing Medical University, Nanjing, China,
2 Department of Urology, The Second Clinical Medical College of Nanjing Medical University, Nanjing, China

☯ These authors contributed equally to this work.
* Weizq11@163.com; Baixinshen@njmu.edu.cn

**Data availability statement:** All relevant data are within the paper and its Supporting Information files.

## Abstract

Interstitial cystitis/bladder pain syndrome (IC/BPS) plagues patients and clinicians with its unclear etiology and pathogenesis, and ineffective treatments. Destruction of epithelial tissue and proliferation of interstitial tissue are typical pathological features of IC/BPS, in which epithelial-mesenchymal transition (EMT) may play an important role. Both the increased urination frequency observed in mice with acute cystitis induced by cyclophosphamide (CYP) and the disruption of the anti-leakage barrier in urothelial cells induced by LPS are associated with the occurrence of EMT. The expression of heparanase 1 (HPSE) and syndecan-2 (SDC-2) is up-regulated in the bladder mucosa of patients with IC, and both of them can promote the development of EMT. Improvement of lower urinary tract symptoms and restoration of the uroepithelial cell anti-leakage barrier in mice with CYP-induced cystitis after treatment with the HPSE inhibitor OGT2115 and inhibited the development of EMT. We then verified that HPSE binds to SDC-2 and that SDC-2 is a key intermediate protein in the pro-EMT role of HPSE, and that EMT was inhibited by knockdown of SDC-2. SDC-2 exerts its biological function by inhibiting the ubiquitinated degradation of TGF-$\beta$R1. Here we identified a novel mechanism by which the HPSE/ SDC-2 axis promotes EMT development and thus causes epithelial dysfunction and altered voiding behavior, providing a new direction for the treatment of IC/BPS.

## Introduction

Interstitial cystitis/bladder pain syndrome (IC/BPS) is a chronic, non-infectious inflammatory disease of the bladder characterized by lower urinary tract symptoms such as urinary frequency and urgency, combined with pain in the pelvic area, bladder or perineal area [1, 2]. The prevalence of IC/BPS ranges from approximately 2.7% to 6.5%, with women being affected at a rate ten times higher than men [3,4]. The global prevalence of IC/BPS is approximately 300 per 100,000 women, with the prevalence among men being about 10%-20% of that in women [5]. When considering only the presence of symptoms suggestive of IC/BPS,

**Funding:** This work was supported by the National Natural Science Foundation of China (82270817, B.X.S.) and the National Natural Science Foundation of China (82370781, Z.Q.W).

**Competing interests:** The authors have declared that no competing interests exist.

the incidence rates for both men and women may increase by more than 10 times [1]. Given the complexity and non-specificity of the clinical signs and symptoms of IC/BPS, and the lack of uniform diagnostic criteria in all regions of the world, the incidence of IC/BPS may be underestimated [6]. Currently, the treatment of IC/BPS mainly focuses on controlling pain and relieving symptoms, and there is no effective and long-lasting treatment, which leads to uncertainty in the prognosis of IC/BPS.

In recent years, the function of the uroepithelium in IC/BPS has been gradually emphasized [7,8]. The role of epithelial-mesenchymal transition (EMT), a standard epithelial pathological change in IC/BPS, has still not been clearly articulated [9]. EMT refers to transforming epithelial cells into cells with a mesenchymal phenotype through a specific program [10]. During this process, epithelial cells exhibit morphological and functional changes, including loss of cell polarity, cytoskeletal alterations, etc [11]. EMT plays a crucial role in developmental processes and is involved in tissue healing, organ fibrosis, and other processes [12]. These biological processes are associated with epithelial dysfunction in IC/BPS and inhibition of EMT occurring in IC/BPS patients may be a new idea for treating IC/BPS [9,13,14].

We have found that heparanase and syndecan-2 (SDC-2) were significantly up-regulated in tissues from IC/BPS patients. Heparanase is an endogenous $\beta$-D-glucuronidase widely found in the extracellular matrix of various tissues and organs [15]. heparanase cleaves the side chain of heparin sulfate and is involved in the degradation and remodeling of the extracellular matrix (ECM), including the release of growth factors such as fibroblast growth factor-2 (FGF-2) [16]. There are two forms of heparanase: heparanase-1 (HPSE) and heparanase-2, with heparanase-1 exerting the primary biological functions, which our study focused on. Several studies have shown that HPSE can promote EMT in bronchial epithelial cells, and renal tubular epithelial cells [17–19]. The multiligand proteoglycan SDC, a family of cell-surface heparan sulfate proteoglycans (HPSG) members, of which four have been identified (SDC-1 to 4), plays an essential function in cell adhesion [20,21]. SDC-2 can promote EMT in colorectal cancer cells, which results in epithelial cells losing cell polarity and their connection to the basement membrane, thereby promoting tumor metastasis and invasion [22,23]. We therefore hypothesized that the HPSE/SDC-2 axis promotes EMT in IC/BPS and aggravates bladder dysfunction. Inhibition of the HPSE/SDC-2 axis restores EMT-induced epithelial dysfunction.

## Materials and methods

### Mice

The experimental animals used in this study were provided by the Animal Center of Nanjing Medical University, weighing 18-20 g, 8-10 weeks old, female C57BL/6J mice. All mice lived in an environment with a light-dark cycle of 12 h, a temperature of 20-26°C, and a relative humidity of 40-60%. Mice were able to eat and drink freely. Under isoflurane anesthesia, mice were executed by cervical dislocation. All experimental procedures were approved by the Animal Ethics and Welfare Committee of Nanjing Medical University (IACUC-2210019).

The mice were divided into three groups: control group, cyclophosphamide (CYP) + vehicle group and CYP+OGT2115 group. The mouse cystitis model was established by intraperitoneal injection of 150 mg/kg of CYP at a concentration of 15 mg/ml. The control group was injected intraperitoneally with an equal dose of saline. Twenty-four hours after the injection of CYP, 5mg/kg of OGT2115 at a concentration of 0.5mg/ml or an equal dose of solvent was injected intraperitoneally into the mice, and the behavioral experiments were performed, or bladder tissues were obtained 24 hours later.

## Voiding spot assays (VSA)

After the mice were acclimatized to their environment, they were gently removed from the rearing cage and placed into a circular metabolic cage with a circular filter paper on the bottom [24]. The room was dark and quiet, and the mice could eat and drink freely during the experiment, which lasted for two hours. The filter paper was obtained after the experiment, dried and placed in an exposure apparatus for imaging. Images were analyzed using Image J software. Suspicious urine stains, such as feces and scratches, were excluded.

## Von frey filament

After the mice were acclimatized, they were placed individually in Plexiglas cubicles with small holes at the bottom. We selected 0.008g, 0.07g, 0.4g, and 1g, 4 fiber filaments for the experiment. Starting with 0.008g, each fiber was stimulated 10 times, and the total score was recorded [25]. The scoring criteria were as follows: 0 points: no response to stimulation of the lower abdomen; 1 point: licking, scratching, escaping, biting and other behaviors appeared after stimulation of the lower abdomen; 2 points: jumping stress action appeared after stimulation of the lower abdomen.

## Urodynamic evaluation

Under 2% isoflurane continuous anesthesia, mice were fixed in the supine position, and the abdominal cavity was opened layer by layer from the middle of the lower abdomen to expose the bladder, which was then accessed by puncture using a scalpel needle and fixed [26]. Saline was injected into the bladder at a rate of 1 ml/h at room temperature to simulate urination. After the mice developed stable micturition, the systolic maximum bladder pressure and the time between contractions were recorded.

## Histopathological analysis

Fresh bladder tissues were obtained from mice after modeling, and the tissues were washed 3 times with PBS to remove blood, urine, etc. The tissues were immersed in 4% paraformaldehyde for 24h for tissue fixation. Paraffin was used for tissue embedding, and the tissue was sliced into 4-6 μm for subsequent experiments. Sections were stained with hematoxylin and eosin (H&E) and photographed under a microscope.

## Cell culture and treatment

SV-HUC-1 was purchased from Zhongqiao Xinzhou Biotechnology. Cells were cultured in a cell culture incubator with a 5% concentration of $CO^2$ at a constant temperature of 37°C. The culture medium was F12K complete medium (89:10:1 ratio of F12K basal medium, serum, and penicillin-streptomycin solution) configured proportionally. The medium was changed once every 2-3 days, depending on the cell status, and all cell manipulations were also done on an ultra-clean bench. At about 70-80% cell confluence, we used LPS 1 μg/ml for 24 h of continuous stimulation to induce EMT and exogenously added OGT2115 5 μmol/L for treatment.

## Protein extraction

Bladder tissue was added to the configured lysate in the ratio of 20 mg tissue: 200 μl RIPA lysate: 2 μl protease inhibitor. Use ophthalmic scissors to cut the epithelial tissue and add steel beads. The tissue was homogenized on a tissue grinder at 4°C for about 5 min until the naked

eye could see no apparent tissue mass. The finished sample was mixed upside down and lysed in the refrigerator for half an hour at 4°C. The lysed sample was centrifuged at 12000rpm at 4°C for 10min, the residue was discarded, and the supernatant was aspirated as a protein stock solution.

The six-well plate full of cells was placed on ice and washed twice with PBS to remove floating cells and residual complete medium, and then the liquid was discarded in the wells. Add about 200 μL of pre-configured lysate to each well. Scrape the cells well on ice using a cell scraper. Pipette the liquid from the wells into a 1.5 ml EP tube and lysed on a shaker at 4°C for 10 min. Centrifuge the supernatant as described above.

After extracting the protein stock solution, the concentration was determined with the BCA quantification kit, adjusted to a homogeneous concentration by adding up-sampling buffer and ddH2O and stored in a refrigerator at -20°C.

## Western blot (WB)

Proteins of different molecular weights were separated using SDS-PAGE gel electrophoresis at constant pressure, and the separated proteins were transferred to a PVDF membrane with constant flow. After blocking for 30 min at room temperature using a rapid blocking solution, the membranes were incubated overnight at 4°C with primary antibody after washing with TBST twice. The next day, the membrane was incubated with horseradish peroxidase-conjugated secondary antibody for 1-2h at room temperature after washing with TBST three times and three more washes with TBST solution. Antibody-antigen complexes were detected using an ECL substrate and visualized with an imaging system.

Primary antibodies are as follows: HPSE Polyclonal antibody (Proteintech), Beta Actin Monoclonal antibody (Proteintech), E-cadherin Polyclonal antibody (Proteintech), Vimentin Polyclonal antibody (Proteintech), N-cadherin Polyclonal (Proteintech), antibody Anti-Syndecan 2/HSPG (Abcam), CD362/SDC2/Syndecan-2 antibody (Santa Cruz), HPA1 (B-4) (Santa Cruz), DYKDDDDK tag Polyclonal antibody (Proteintech), 6*His, His-Tag Monoclonal antibody (Proteintech) Smad2/3 Antibody (Affinity), Phospho-Smad2/3 (Thr8) Antibody (Affinity), ubiquitin Polyclonal antibody (Proteintech).

## Immunohistochemistry (IHC) and immunofluorescence (IF)

Prepare animal tissue sections or cell crawls in advance. Submerge samples with drops of BSA for 30 minutes, then incubate overnight at 4°C with primary antibody. After washing, the samples were incubated with a secondary antibody for 1h at room temperature, protected from light, and then the residual liquid was washed away. Immunohistochemistry was performed using 3,3'-diaminobenzidine staining followed by hematoxylin re-staining of nuclei. Immunofluorescence was performed by staining nuclei with 4',6-diamidino-2-phenylindole, followed by autofluorescence quenching.

## Epithelial permeation assay

SV-HUC-1 cells were inoculated in the upper layer of a 24-well Transwell plate and cultured for 24h to allow them to form a monolayer cell barrier. Different drugs were added to treat the cells for 24h, followed by washing away the residual drugs. 500μl of 1mg/ml FITC-dextran solution was added to the upper layer, and HBSS was added to the lower layer. 50μl of HBSS in the lower chamber was pipetted to a 96-well plate shielded from light after incubation at 37°C and protected from light for 1h. The relative fluorescence intensity was measured and recorded at the excitation wavelength of 485nm and the emission wavelength of 530nm.

## Immunoprecipitation (IP)

Cells were inoculated in Petri dishes to about 80% confluence and treated with the addition of different drugs for 24 h. The medium was discarded and washed twice with PBS. Add 0.5-1 ml of pre-configured IP lysate (containing protease inhibitors) to the culture dish, and the cells were scraped by cell scraping. The scraped cells were transferred to EP tubes and lysed on ice for 30 min. Centrifuge the cells at 12000 rpm for 10 min at 4°C and collect the supernatant. Aspirate the magnetic beads and wash the beads 3 times. Resuspend the beads with 100 μl of washing solution, add the desired amount of primary antibody, and shake and mix at room temperature for 30 min to bind the beads. Discard the primary antibody, wash 3 times, add the supernatant, and shake at 4°C overnight to prevent the beads from settling. The supernatant was discarded the next day, the magnetic beads were washed 3 times, and 1X loading buffer 60μl was added to resuspend the magnetic beads. The beads were heated at 70°C for 10 min and immediately placed on a magnetic rack, and the upper layer of liquid was aspirated and stored at -20°C in a refrigerator.

## Data analysis

Measurement data were expressed as mean $\pm$ standard deviation ('x $\pm$ SD), and t-test or one-way ANOVA was used for intergroup comparison, and Tukey's method was used for two-by-two comparisons; within-group comparisons of continuously measured indexes were performed by repeated-measures ANOVA, and comparisons between groups at each time point were performed by two-way ANOVA and correlation analyses were performed by Pearson's or Spearman's methods, and a $P<0.05$ was considered as the difference being statistically significant. The sample size and specific statistical methods were adjusted accordingly.

## Results

### EMT correlates with urination frequency in mice and barrier function in urothelial cells

To investigate the correlation between the frequency of urination in mice and epithelial-mesenchymal transition (EMT), we quantified the number of urinations by counting the urine spots on filter paper in the voiding spots experiment. Additionally, we conducted a Western blot (WB) assay to semi-quantitatively analyze EMT-related proteins as a measure of EMT progression. Following intraperitoneal injection, cyclophosphamide (CYP) is metabolized in the kidney into acrolein, a toxic compound to urothelial cells. This process induces urothelial injury and inflammation, ultimately triggering EMT. Mice in the cyclophosphamide (CYP) group had smaller and more urine spots on the filter paper, suggesting that the frequency of urination was significantly increased and the urine volume was reduced (Fig 1A). Western blot (WB) results showed that the expression of the epithelial marker E-cadherin was decreased and the expression of the mesenchymal markers N-cadherin and Vimentin was increased (Fig 1B and 1C). We plotted the correlation curve between EMT and the number of urination, and the correlation coefficients showed that the occurrence of EMT correlated with increased urination in mice (Fig 1D). This result demonstrates a correlation between the occurrence of EMT and urination frequency in mice, indicating that the progression of EMT increases urination frequency in mice.

We then constructed an urothelial EMT model in vitro by continuously stimulating Sv-huc-1 with lipopolysaccharide (LPS) for 24 hours to further investigate the relationship

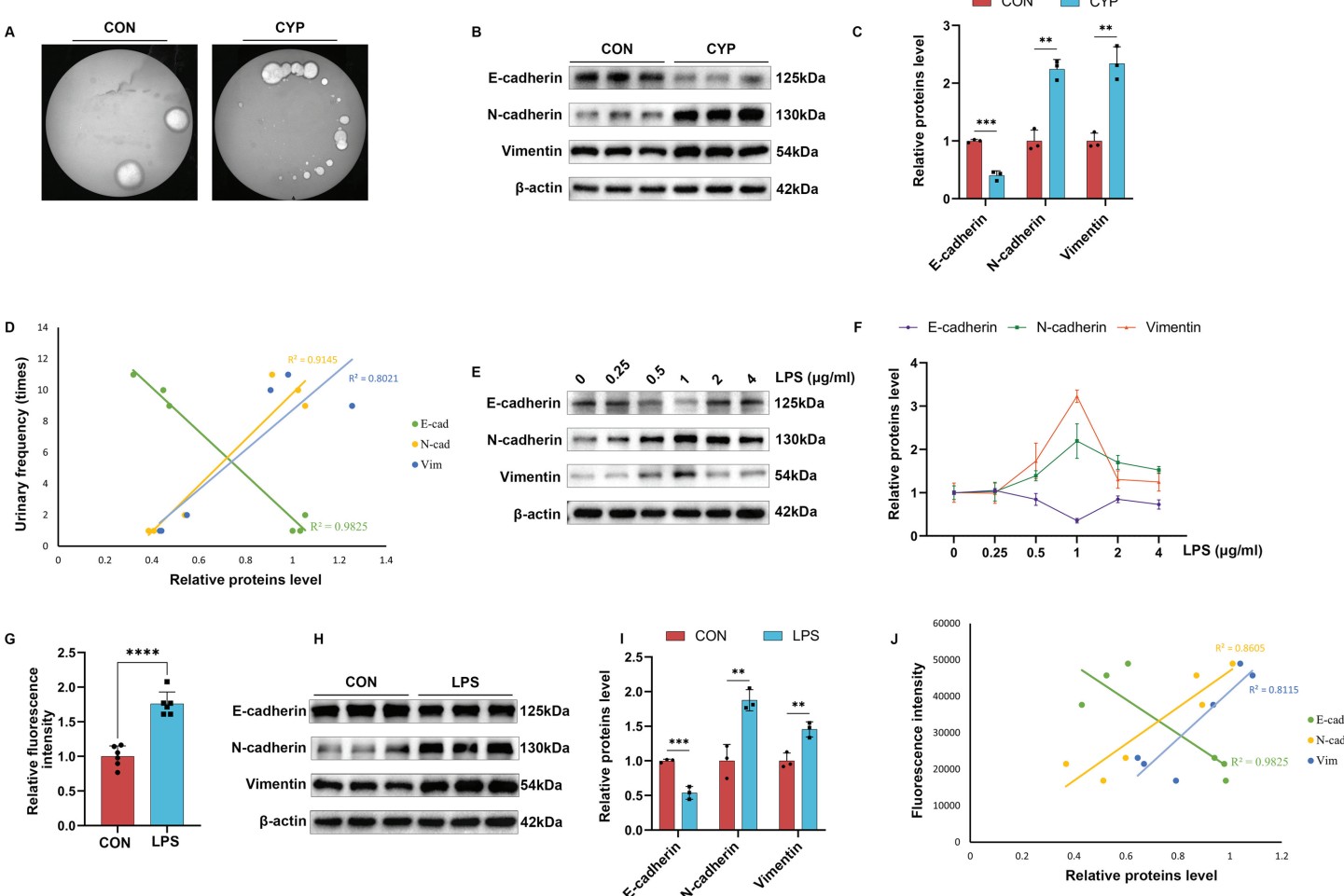

**Fig 1. Epithelial-mesenchymal transition (EMT) correlates with urination frequency in mice and barrier function in urothelial cells.** A: Results of voiding spots of control (CON) and cyclophosphamide (CYP) groups. B-C: Western blot (WB) results and semi-quantitative analysis of EMT indexes of bladder proteins in mice of CON and CYP groups (n=3). D: Correlation curves of EMT with the number of voiding. E-F: WB results and semi-quantitative analysis of EMT index of cellular proteins in CON and lipopolysaccharide (LPS) groups. G: Relative fluorescence intensity of leakage occurring in CON and LPS groups(n=6). H-I: WB results and semi-quantitative analysis of cytosolic protein EMT indexes in CON and LPS groups(n=3). J: Correlation curve of EMT with fluorescence intensity of leakage. Statistical analyses were performed using t-tests with error lines as mean ± standard error, **P < 0 .01, ***P < 0 .001, ****P < 0.0001.

between EMT and urothelial cell function. The epithelial permeability assay was employed to evaluate the barrier function of urothelial cells, where the fluorescent intensity of leakage is inversely proportional to the barrier function. A concentration gradient of LPS 0, 0.25, 0.5, 1, 2, and 4 µg/ml was set up to explore the most suitable intervention concentration of LPS and to make a dose-effect curve. The results showed that the most significant EMT occurred in cells at the LPS intervention concentration of 1 µg/ml. (Fig 1E and 1F). To verify the altered epithelial cell function, we performed epithelial permeability experiments. The results showed that more fluorescein (FITC) fluorescence leakage occurred under LPS treatment, suggesting that LPS treatment leads to disruption of epithelial function (Fig 1G). We confirmed that EMT occurred in epithelial cells under LPS 1µg/ml intervention for 24 h (Fig 1H and 1I). We plotted the correlation curve between EMT and fluorescence leakage, and the correlation

coefficients showed that EMT occurrence was correlated with urothelial cell function (Fig 1J). This result demonstrates a correlation between the occurrence of EMT and urothelial cell barrier function, suggesting that the progression of EMT disrupts the epithelial cell barrier function. Taken together, these experimental findings indicate that EMT is associated with urination frequency in mice and barrier function in urothelial cells.

## OGT2115 inhibits HPSE to improve voiding behavior in cystitis mice

To answer this question, we employed bioinformatics analysis combined with WB to screen for differentially expressed genes HPSE and SDC-2. Additionally, we used the HPSE inhibitor OGT2115 to treat CYP-induced cystitis in mice and validate phenotypic changes. The interstitial cystitis dataset GSE57560 was downloaded from the GEO public database, containing 12 interstitial cystitis and 4 control bladder biopsy tissues from humans. We found that heparanase 1 (HPSE) and syndecan-1 (SDC-1), syndecan-2 (SDC-2) and syndecan-4 (SDC-4) within the SDC family were up-regulated in the interstitial cystitis/bladder pain syndrome (IC/BPS) tissues (Fig 2A and 2B). The expression of SDC-1 and SDC-4 in the bladders of CYP-induced cystinosis mice was low and not significantly different, and the expression of SDC-2 and HPSE was significantly elevated in the CYP group (Fig 2C). Therefore, the HPSE/SDC-2 axis was selected for subsequent experiments.

To further calculate the volume of urination in mice, we dropped a fixed volume of urine and measured its area to fabricate an area-volume standard curve (Fig 2D). OGT2115, as an HPSE inhibitor, was able to ease the behaviors of increased frequency of urination and decreased average volume of urination occurring in CYP-induced cystitis mice. The total volume of urination was not statistically different from that of the CYP group (Fig 2E and 2F). Lower abdominal sensitivity increased and pain score was elevated in the CYP group, and lower abdominal pain was relieved after OGT2115 treatment (Fig 2G). The urodynamic results showed that the urine storage period was significantly shortened, and the maximal forced urethral muscle contraction decreased in the CYP group, and the mice had a longer voiding interval and an elevated maximal forced urethral muscle contraction after OGT2115 treatment, which restored the micturition behavior to a near-normal level (Fig 2H and 2I). This shows that the HPSE inhibitor OGT2115 was able to ameliorate the urinary symptoms of increased voiding frequency and decreased urine output in mice with CYP-induced acute cystitis and reduce lower abdominal sensitivity.

## OGT2115 inhibits EMT in the bladders of mice with CYP-induced cystitis

We employed WB combined with immunohistochemistry (IHC) to observe the expression and distribution of EMT-related proteins, while HE staining was used to observe morphological changes in bladder tissue. At the in vivo level by WB assay, OGT2115 was shown to down-regulate the expression of HPSE and SDC-2 (Fig 3A). Then we examined the expression levels of EMT-related proteins. The results indicated that in the OGT2115 group, the expression of E-cadherin was increased compared to the CYP group, while the expression levels of N-cadherin and Vimentin were decreased, returning to levels close to those observed in the CON group (Fig 3B and 3C). Hematoxylin-eosin (HE) staining results of bladder tissues of mice showed that the CYP group showed a decrease in the number of uroepithelial cell layers and subepithelial tissue edema, and OGT2115 perfusion was able to restore the epithelial coverage and reduce the subepithelial tissue edema (Fig 3D). IHC results showed that OGT2115 was able to restore E-cadherin expression, reduce Vimentin distribution, and bring them back to levels similar to those observed in the CON group (Fig 3E-G). The above experimental

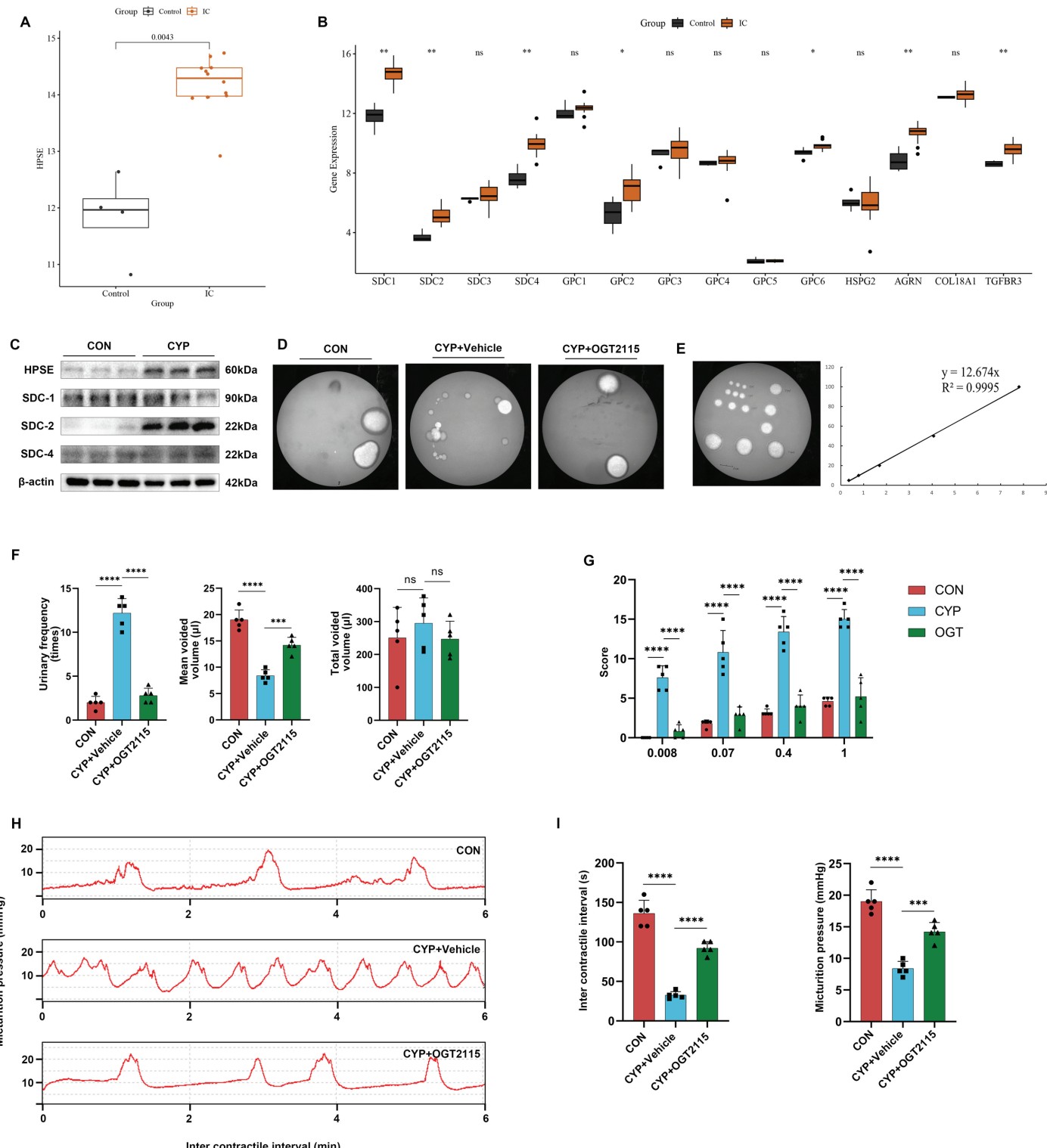

**Fig 2. OGT2115 inhibits HPSE to improve voiding behavior in cystitis mice.** A-B: Bioinformatics analysis of HPSE and HPSGs using the GSE57560 dataset. C: WB results of HPSE and syndecan (SDC) family proteins in the bladder of mice in the CON and CYP groups. D: Results of voiding spots of OGT2115-treated mice. E: Fixed volumes (5, 10, 20, 50, 100 $\mu$l) of mouse urine were dropped onto filter paper for experimental use. ImageJ was used to calculate the area, and an area-volume standard curve was plotted, resulting in the formula: Volume = Area * 12.674. F: Based on the area of mouse urine spots, quantitative analysis was conducted on the frequency of urination, average urine volume, and total urine volume in mice (n=5). G: Pain scores measured by von frey filament (n=5).H-I: Urodynamic measurements in mice, statistical analyses of micturition intervals, and maximal forcing muscle contraction force (n=5).Statistical analyses were performed using one-way ANOVA with error lines of mean ± standard error, ***P < 0 .001, ***P < 0.0001.

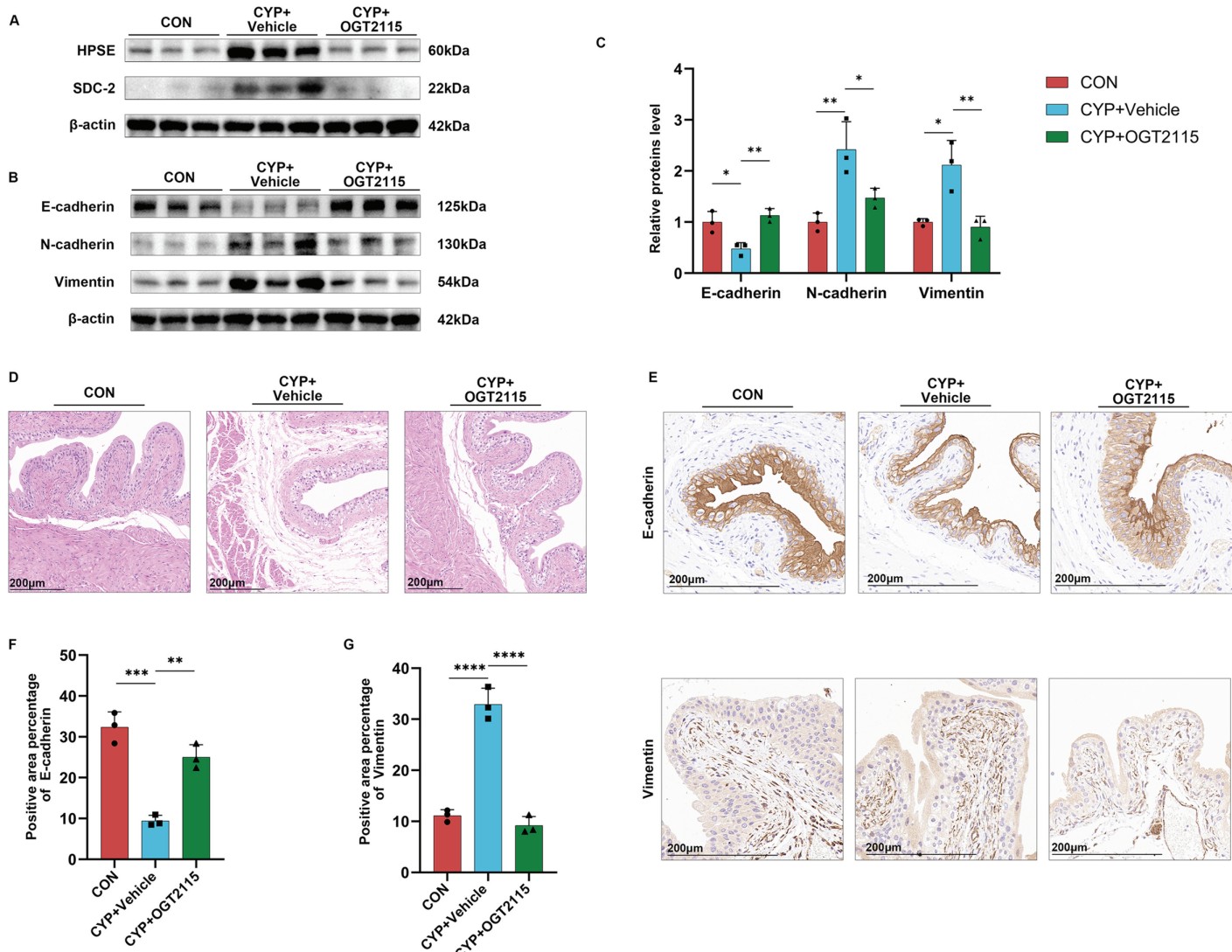

**Fig 3. Inhibition of HPSE ameliorates the EMT in mice's bladder with CYP-induced cystitis.** A: WB results of HPSE and SDC-2 in the bladder of mice after OGT2115 treatment. B-C: WB results of EMT indexes of bladder proteins in the bladder of mice after OGT2115 treatment and semi-quantitative analyses (n=3). D: Hematoxylin-eosin (HE) staining of the bladder of mice after OGT2115 treatment. E-G: Immunohistochem ical results of E-cadherin and Vimentin in the bladder of OGT2115-treated mice and statistical analysis of positive area (n=3). Statistical analyses were performed using one-way ANOVA with error lines of mean ± standard error, *P < 0.05, **P < 0 .01, ***P < 0 .001, ****P < 0.0001.

results suggested that OGT2115 was able to inhibit EMT, which may be an important process to improve the behavior of mice.

## OGT2115 suppresses LPS-induced EMT in Sv-huc-1

We added LPS stimulation to construct a cellular EMT model and intervened by adding OGT2115 to the culture medium. WB and immunofluorescence (IF) were used to observe the expression of EMT-related proteins, and urothelial permeation assay was employed to verify the urothelial barrier function. To explore the optimal intervention concentration of OGT2115, we established a concentration gradient of OGT2115 at 0, 1.25, 2.5, 5, 10, and

20 μmol/L. Initially, we used the CCK8 method to measure the cytotoxicity of OGT2115, with the control group receiving DMSO as the solvent. After CCK8 incubation, we measured the OD values, subtracted the blank well value (only culture medium added) from each well, and divided by the control group to obtain the percentage of live cells. After importing the data into Prism, we plotted a fitted curve using the logarithm of concentration as the x-axis and cell survival rate as the y-axis. From this, we calculated an IC50 value of 36.26 μM (Fig 4A). Following this, we conducted a WB experiment and generated a dose-response curve. The results showed that when the OGT2115 intervention concentration was 5 μmol/L, the expression of E-cadherin was the highest, and the expression of N-cadherin, and Vimentin was the lowest, indicating that at this time, the inhibitory effect of OGT2115 on EMT was the most significant, so we chose 5 μmol/L for the subsequent experiments (Fig 4B and 4C). OGT2115 at this concentration inhibited LPS-induced activation of HPSE and SDC-2 (Fig 4D). The LPS-induced decrease in E-cadherin expression and increase in N-cadherin and Vimentin expression was restored under OGT2115 treatment (Fig 4E and 4F). The results of IF were consistent with the results of WB (Fig 4G-I). Epithelial permeability assay showed that more FITC leakage occurred in the LPS group, and OGT2115 restored the anti-leakage function of uroepithelial cell formation (Fig 4J). This shows that OGT2115 was able to inhibit LPS-induced EMT in uroepithelial cells and restore their anti-leakage function.

## SDC-2 is a crucial intermediate protein of HPSE-induced EMT

To answer this question, we employed multiple IF (mIF) and co-immunoprecipitation (CO-IP) techniques to verify whether HPSE binds to SDC-2 and conducted SDC-2 knockdown to confirm it as a downstream protein. For investigating the relationship between SDC-2 and HPSE, we found with the help of mIF that HPSE was significantly activated with SDC-2 in the LPS group, the fluorescence intensity was enhanced, and the expression sites of HPSE and SDC-2 were highly overlapped (R=0.93) (Fig 5A). To further explore whether SDC-2 could bind to HPSE, we performed CO-IP experiments, which showed that HPSE bound to SDC-2 (Fig 5B). To clarify the role played by SDC-2, we transfected the siRNA of SDC-2 for knockdown experiments. By verifying the knockdown efficiency, the most efficient SDC-2-si-2 was selected for subsequent experiments (Fig 5C). By analyzing the expression of E-cadherin, N-cadherin, and Vimentin through WB experiments, we found that the EMT transformation occurred in epithelial cells in the LPS group was restored after knockdown of SDC-2 (Fig 5D and 5E). IF results showed that after the knockdown of SDC-2, E-cadherin expression and distribution were restored, and Vimentin expression and distribution decreased after the knockdown of SDC-2 (Fig 5F-H). We further used an epithelial permeability assay and found that the knockdown of SDC-2 improved LPS-induced epithelial cell dysfunction and reduced FITC leakage (Fig 5I).

Due to the potential additional biological effects of inhibitors, we overexpressed HPSE to observe the biological function of this protein in EMT. We overexpressed HPSE and validated using WB (S1A Fig). Overexpression of HPSE promoted EMT progression, epithelial dysfunction, and increased fluorescence leakage occurred (S1B-D Fig). We knocked down SDC-2 on the basis of overexpression of HPSE (S1E Fig). The EMT-promoting effect exerted by overexpression of HPSE was counteracted by SDC-2 knockdown (S1E and S1F Fig). Epithelial function was restored and fluorescence leakage was reduced after SDC-2 knockdown (S1G Fig). Combined with the previous results, it can be concluded that HPSE binding to SDC-2 caused downstream EMT development and epithelial dysfunction, with SDC-2 playing a pivotal role.

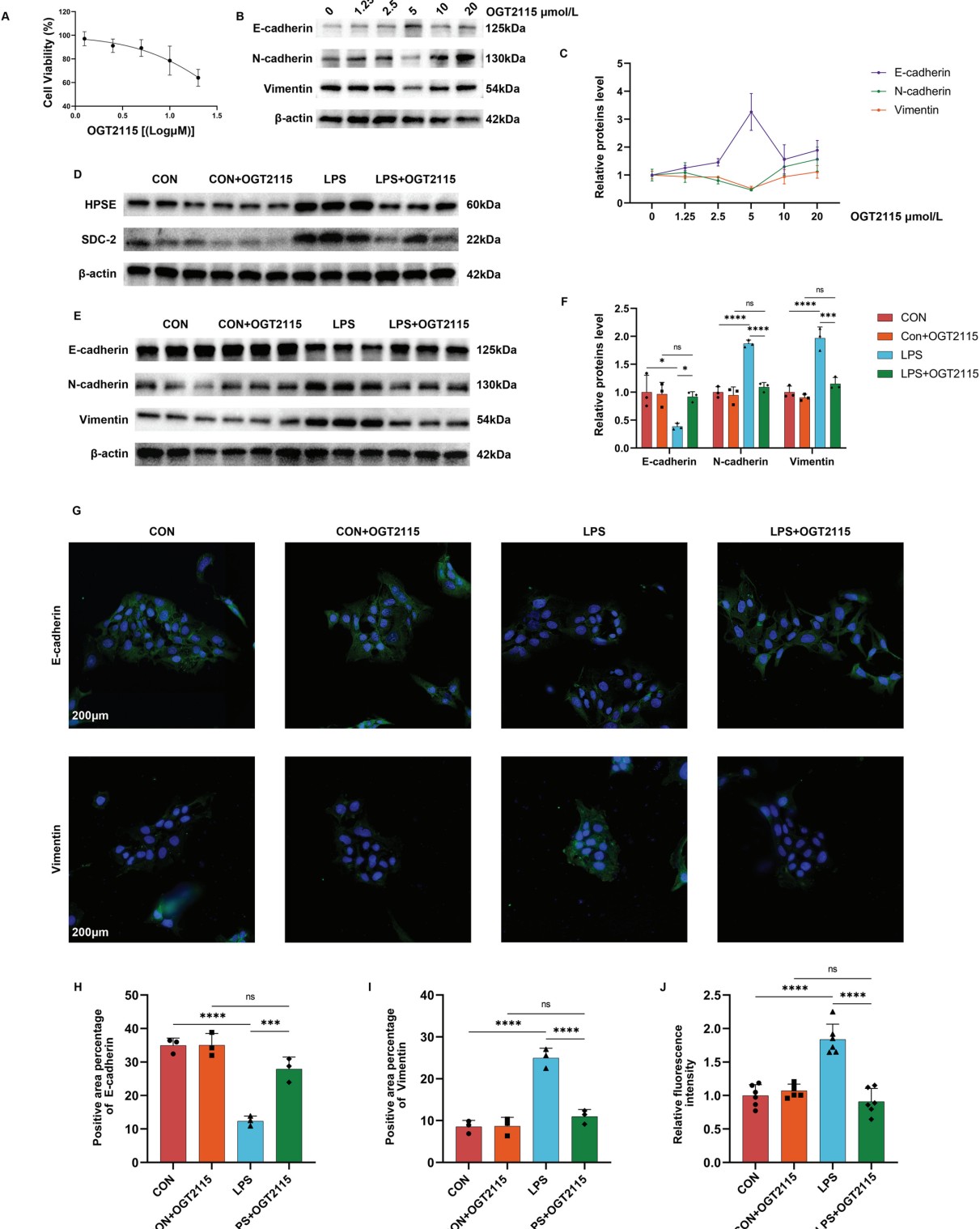

**Fig 4. Inhibition of HPSE improves LPS-induced EMT of Sv-huc-1 occurrence.** A: Cytotoxicity of OGT2115 was determined using the CCK8 assay (n=5). B-C: WB results and dose-effect curves of EMT indexes under the concentration gradient of OGT2115. D: WB results of HPSE and SDC-2 of cells after OGT2115 treatment. E-F: WB results of EMT indexes of cellular proteins after OGT2115 treatment and semi-quantitative analysis (n=3). G-I: Immunofluorescence results of E-cadherin and Vimentin in OGT2115-treated cells and statistical analysis of positive area (n=3). J: Relative fluorescence intensity of leakage occurred after OGT2115 treatment (n=6). Statistical analyses were performed using one-way ANOVA with error lines of mean ± standard error, *P < 0.05, ***P < 0.001, ****P < 0.0001.

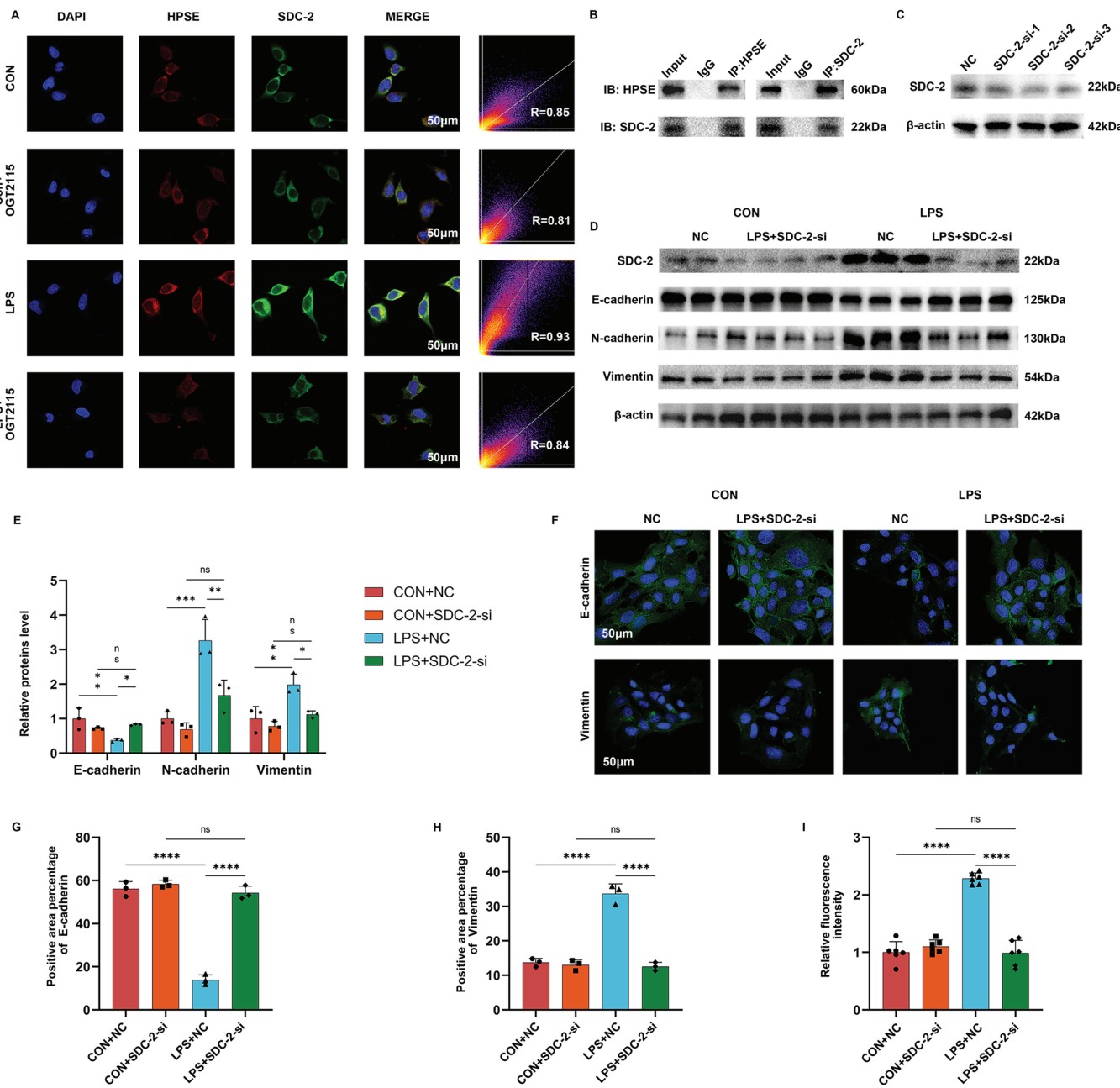

**Fig 5. SDC-2 is a downstream protein of HPSE and plays an essential role in EMT.** A: Multiplex immunofluorescence results of HPSE and SDC-2. B: CO-immunoprecipitation (CO-IP) results of HPSE and SDC-2. C: Validation of the knockdown efficiency of SDC-2-siRNA. D-E: WB results of the EMT indexes of the cellular proteins and semi-quantitative analysis of the cellular proteins after SDC-2 knockdown (n=3). F-H: Immunofluorescence results of E-cadherin and Vimentin in the cells after SDC-2 knockdown and statistical analysis of their positive areas (n=3). I: Relative fluorescence intensity of the leakage that occurred after SDC-2 knockdown (n=6). Statistical analyses were performed using one-way ANOVA with error lines of mean $pm$ standard error, *P < 0.05, **P <0 .01, ***P <0 .001, ****P <0.0001. NC: Negative control.

## SDC-2 inhibits TGF-$\beta$1 ubiquitination degradation and thereby activates the TGF-$\beta$ pathway

The above experiments demonstrate that SDC-2 is a downstream protein that binds to HPSE, but the specific binding site remains unclear. Therefore, based on the HS chain region of SDC-2, we designed truncations (Fig 6A). Transfection of SDC-2, D1, and D2 with Flag tag was

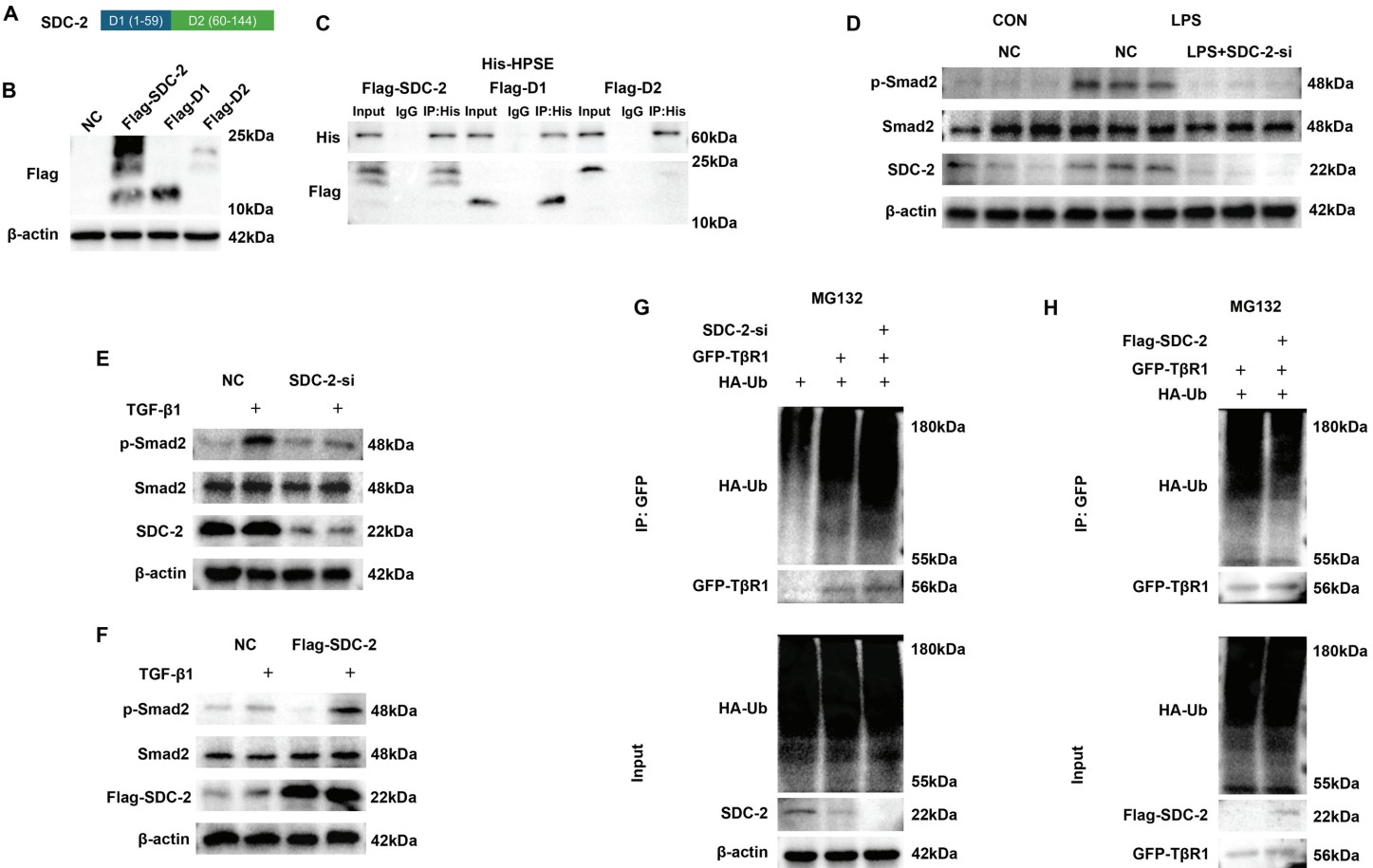

**Fig 6. SDC-2 inhibits TGF-$\beta$1 ubiquitinated degradation thereby activating the TGF-$\beta$ pathway.** A: Amino acid sequence of SDC-2 truncosome. B: WB validation of transfected Flag-SDC-2, Flag-D1, Flag-D2. C: CO-IP of transfected His-HPSE with Flag-SDC-2, Flag-D1, Flag-D2 results. D: WB results of TGF-$\beta$ pathway after SDC-2 knockdown. E: Pathway protein validation of exogenously added TGF$\beta$1 (100pM) after SDC-2 knockdown. F: Pathway protein validation of exogenously added TGF$\beta$1 (100pM) after overexpression of SDC-2. G: ubiquitylation level of TGF-$\beta$R1 after knockdown of SDC-2.H: Pathway protein validation of exogenously added TGF$\beta$1 (100pM) after overexpression of SDC-2. 2 after ubiquitination level of TGF-$\beta$R1.NC: Negative control.

verified using WB (Fig 6B). HPSE has the function of binding to HS chain, so we used HPSE pull-down protein with His tag to verify the binding site with SDC-2. The CO-IP results showed that HPSE binds to D1, but not to D2 (Fig 6C).

Transforming growth factor-$\beta$ (TGF-$\beta$) is a profound factor that promotes fibrosis and EMT, and it is also a downstream signaling pathway of HPSE. When TGF-$\beta$1 binds to its receptor, TGF$\beta$R1, it can phosphorylate the downstream protein SMAD2. Phosphorylated SMAD2 can bind to SMAD4 and enter the nucleus, activating the signaling pathway through the canonical pathway. We found that it was significantly activated in the LPS intervention group and inhibited after knockdown of SDC-2 (Fig 6D). Then we exogenously added TGF-$\beta$1 as a pathway activator, and we found that the activation of the pathway by TGF-$\beta$1 disappeared after SDC-2 knockdown (Fig 6E). And the activating effect of TGF-$\beta$1 was further enhanced after overexpression of SDC-2 (Fig 6F). The above results show that even with exogenous addition of TGF-$\beta$1, the TGF$\beta$ signaling pathway cannot be significantly activated after knocking down SDC-2, while the activation effect of TGF-$\beta$1 is significantly enhanced after overexpressing SDC-2. This result suggests that the activating effect of SDC-2 on the

TGF-$\beta$ pathway is not through the promotion of TGF-$\beta$1 secretion, but may exert its biological function through its receptor. Ubiquitination is one of the pathways that promote protein degradation, and we examined the level of ubiquitination of TGF-$\beta$ receptor 1 (TGF-$\beta$R1) under knockdown and overexpression of SDC-2. The results showed that the ubiquitination level of TGF-$\beta$R1 increased after knockdown of SDC-2, and the ubiquitination level of TGF-$\beta$R1 decreased after overexpression of SDC-2 (Fig 6G and 6H). Taken together, we can conclude that SDC-2 can inhibit TGF-$\beta$R1 ubiquitination degradation thereby activating the TGF-$\beta$ pathway.

## Discussion

IC/BPS is a chronic aseptic bladder inflammation characterized by symptoms of urinary frequency, urgency, pelvic and bladder pain, and poor outcome, which is a major clinical challenge [27,28]. The pathogenesis of IC/BPS is currently unclear, while recent studies have suggested that epithelial dysfunction and bladder fibrosis due to EMT may be the initial lesion of IC/BPS.

The bladder surface is covered with a cellular barrier composed of umbrella, intermediate, and basal cells. This prevents urine and toxic substances from leaking into the submucosa and causing tissue inflammation [29,30]. EMT can be oncogenic in tumors, disrupt epithelial function, and promote tissue fibrosis in non-neoplastic diseases [31]. EMT disrupts this cellular barrier, which triggers chronic inflammation of the subepithelial bladder, leading to bladder dysfunction [32,33]. We observed a positive correlation between the number of micturitions and the occurrence of EMT by intraperitoneal injection in a mouse model of CYP-induced cystitis. We induced EMT in Sv-huc-1 using LPS in vitro and found that epithelial cell barrier leakage was positively correlated with EMT.

HPSE and its downstream protein SDC-2 can each promote EMT. HPSE was found to promote the expression of the pro-fibrotic factors transforming growth factor and fundamental fibroblast growth, promoting EMT. In contrast, SDC-2 was found to promote EMT via the MAPK pathway and other pathways in tumor cells to exert oncogenic effects [34, 35]. Although both HPSE and SDC-2 can each play a role in promoting EMT, it is not clear whether they have a synergistic effect. We first found that HPSE and SDC-2 were significantly up-regulated in IC/BPS patients and then found that inhibition of HPSE using OGT2115 resulted in the alleviation of lower urinary tract symptoms in CYP-induced cystitis mice and restored LPS-induced disruption of the leakage barrier of epithelial cells. EMT occurring in CYP and LPS-induced EMT was inhibited after OGT2115 treatment. To further investigate the relationship between SDC-2 and HPSE, we found that HPSE was highly co-localized and had a binding relationship with SDC-2 through IP and multiplex immunofluorescence experiments. The biological function exerted by HPSE disappeared, and EMT was inhibited after the knockdown of SDC-2, suggesting that SDC-2 is a key intermediate protein in causing EMT after HPSE activation. By constructing a truncated body we verified that HPSE binds to the D1 structural domain of SDC-2. The TGF$\beta$ pathway is a downstream pathway of HPSE and is able to promote EMT.SDC-2 is able to inhibit the ubiquitinated degradation of TGF-$\beta$R1 thereby activating the TGF-$\beta$ pathway, a newly identified pathway of action of HPSE.

This study elucidated the correlation between EMT and urination in vivo. The correlation between EMT and epithelial anti-leakage function in vitro was proved, which provided a new pathophysiologic mechanism for IC/BPS. Then, we identified SDC-2, an essential intermediate protein of HPSE to promote EMT, and confirmed the binding relationship between HPSE and SDC-2 using immunoprecipitation and multiplex immunofluorescence. SDC-2

exerts its biological function by inhibiting ubiquitinated degradation of TGF-$\beta$R1.In summary, the present study suggests that inhibition of the HPSE/SDC-2 axis can improve the EMT occurring in IC/BPS, which may be a new target for treating IC/BPS.

## Supporting information

**S1 Fig Knockdown of SDC-2 inhibits EMT activated by overexpression of HPSE.** A: WB validation of transfection of His-HPSE. B: WB results of EMT metrics after transfection of His-HPSE. C: IF results of EMT metrics after transfection of His-HPSE. D: Relative fluorescence intensities of leakage occurring after transfection of His-HPSE (n=6). E: WB results of EMT metrics after transfection of WB results of EMT metrics after transfection of His-HPSE and SDC-2-siRNA. F: Immunofluorescence results of EMT metrics after transfection of His-HPSE and SDC-2-siRNA. G: Relative fluorescence intensity of leakage occurring after transfection of His-HPSE and SDC-2-siRNA (n=6). Statistical analyses between two groups were performed by t-test and between three groups by one-way ANOVA, with error line as mean $\pm$ standard error, ****P < 0.0001. NC: Negative control.
(XLX)

**S1 Table Raw data for bar graphs.**
(PDF)

## Author contributions

**Conceptualization:** Baixin Shen, Zhongqing Wei.
**Data curation:** Zhengsen Chen, Yuting He, Junjie Zhang.
**Formal analysis:** Zhengsen Chen, Yuting He.
**Funding acquisition:** Baixin Shen, Zhongqing Wei.
**Investigation:** Yuting He, Qingyu Ge, Tianpeng Du, Zongyao Fan, Junyi Zhou.
**Methodology:** Zhengsen Chen, Yuting He, Qingyu Ge, Tianpeng Du, Zongyao Fan, Xin Yang.
**Project administration:** Zhongqing Wei.
**Resources:** Zhongqing Wei.
**Software:** Junjie Zhang.
**Supervision:** Baixin Shen, Zhongqing Wei.
**Validation:** Zhengsen Chen, Junjie Zhang, Xin Yang, Zhongqing Wei.
**Visualization:** Yuting He.
**Writing – original draft:** Zhengsen Chen.
**Writing – review & editing:** Yuting He, Junjie Zhang, Baixin Shen, Zhongqing Wei.

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
