## [Decision Letter · Decision Letter 0]

13 Nov 2024

PONE-D-24-45975Inhibition of HPSE/SDC-2 axis-induced epithelial-mesenchymal transition for treating IC/BPSPLOS ONE

Dear Dr. He,

Thank you for submitting your manuscript to PLOS ONE. After careful consideration, we feel that it has merit but does not fully meet PLOS ONE’s publication criteria as it currently stands. Therefore, we invite you to submit a revised version of the manuscript that addresses the points raised during the review process.

As stated in the reviews, you should improve the introduction section and clarify the hypothesis. Additional experiments in deciphering the mechanism should be proposed as well as other markers to characterize the EMT (like TGF-beta). Also, the Results section needs to be better organized as suggested by one of the reviewers.

We look forward to receiving your revised manuscript.

Kind regards,

Boyan Grigorov

Academic Editor

PLOS ONE

“This work was supported by the National Natural Science Foundation of China (82270817, B.X.S.) and the National Natural Science Foundation of China (82370781, Z.Q.W).”

Reviewers' comments:

Reviewer's Responses to Questions

**Comments to the Author**

1. Is the manuscript technically sound, and do the data support the conclusions?

Reviewer #1: Partly

Reviewer #2: Partly

2. Has the statistical analysis been performed appropriately and rigorously? 

Reviewer #1: I Don't Know

Reviewer #2: No

3. Have the authors made all data underlying the findings in their manuscript fully available?

Reviewer #1: Yes

Reviewer #2: Yes

4. Is the manuscript presented in an intelligible fashion and written in standard English?

Reviewer #1: Yes

Reviewer #2: No

5. Review Comments to the Author

Reviewer #1: In their study, Chen et al. describe the involvement of the HPSE/SDC-2 axis in the development of epithelial-to-mesenchymal transition (EMT) in interstitial cystitis/bladder pain syndrome (IC/BPS) and propose a new therapeutic target for this pathology. In vivo and in vitro experiments were conducted to test the small molecule OGT2115, which was confirmed to inhibit the EMT process. The in vitro studies indicate the potential involvement of HPSE and SDC-2 in the development of EMT and suggest that their inhibition may facilitate the restoration of a favorable phenotype. The authors conclude that OGT2215, an HPSE inhibitor, may represent a promising therapeutic molecule for IC/PBS.

I have a few comments/suggestions to improve and strengthen the study:

• The authors used OGT2115, a HPSE-1 inhibitor, to unveil HPSE-1 role on EMT. The choice of 5 µM as an optimal drug concentration was based solely on EMT phenotypic changes (decrease of E-cadherin and increase of Vimentin at protein level). However, Western blots of Fig 4A-B do not show a strong inhibition of HPSE-1 expression at this concentration. What is the IC50 of OGT2115 on HPSE-1 expression?

• It is important to perform cytotoxicity assays of OGT2115 to confirm the non-cytotoxic effect of the drug.

• Title and conclusion of the 3.4 paragraph are not very clear. Is there an improvement or an inhibition of the EMT process under OGT2215 treatment? Clarify your message.

• The authors focused on the HPSE/SDC-2 axis implication by modulating HPSE (through OGT2115) but to strengthen and confirm their observations, the use of SDC-2 inhibitors could be interesting. Similarly, the authors performed RNAi experiments with only SDC-2 RNAi. It would be interesting to genetically modulate HPSE expression (by siRNA), deplete both SDC-2/HPSE and finally rescue their expression, to confirm that both proteins are essential.

• In their study, the authors analyzed GSE57560 dataset and described an upregulation for SDC-2 but also for SDC-1 and -4. However, they did not go further with these receptors because there are not significant changes in mice. Can the authors discuss more on that aspect? What is the difference between human and mice for SDC expression?

• As they use human cell lines, it could be interesting to perform some experiments in vitro on SDC-1 and -4. Do they interact with HPSE? Are they involved in EMT process? Can they partially replace SDC-2 when SDC-2 is knockdown?

• Finally, they only used phenotypic markers to describe the EMT process (E-cadherin as epithelial marker and Vimentin for mesenchymal marker), but they did not control cytokine expression, already described in the literature to be involved in this process (e.g. TGF-β). In the discussion part, the authors suggest that HPSE can activate TGF-β (without citation). Regarding these informations, it could be interesting to add a TGF-β inhibitor to confirm the EMT induction is TFG-β -dependent.

Reviewer #2: The manuscript investigates the role of HPSE (heparanase) and SDC-2 in epithelial-mesenchymal transition (EMT) within the context of interstitial cystitis/bladder pain syndrome (IC/BPS), proposing that the inhibition of the HPSE/SDC-2 axis could mitigate EMT-related epithelial dysfunction. While the study presents intriguing data and utilizes multiple methodologies (in vivo, in vitro), there are critical areas where improvements are necessary :

in the introduction section :

• It might be helpful to add specific epidemiological data on IC/BPS, like prevalence or challenges in diagnosis. Including a bit more detail on current treatments and their limitations could help to show why targeting EMT and HPSE/SDC-2 might offer something new in terms of therapy.

• Adding a few lines on the functions of HPSE and SDC-2, especially their roles in extracellular matrix remodeling and EMT, might help readers unfamiliar with these molecules to understand why they matter in IC/BPS

• The hypothesis is implicit rather than clearly stated. While the introduction discusses the potential involvement of HPSE and SDC-2 in EMT in IC/BPS, it lacks an explicit hypothesis that clearly directs the study's objectives. e.g., “We hypothesize that the HPSE/SDC-2 axis promotes EMT in IC/BPS and that its inhibition can ameliorate EMT-associated epithelial dysfunction.”.

• You should add a few more references specifically for EMT in bladder disease could strengthen your argument. For exemple, citing related literature on EMT in non-cancerous inflammatory conditions would more enhance the study’s novelty.

Methodology and results

• The Results section should be restructured for clarity and logical flow. Each sub-section should begin with a brief statement of specific objectives, followed by a concise summary of the methods used.

• Also, there is sometimes disorganized presentation of findings, for exemple, the abrupt presentation of EMT in bladder tissues, before first confirming cystitis validation, disrupts the logical progression of the results. This sequence should ideally be reversed to establish the model before discussing EMT induction.

• Although the manuscript discusses the HPSE/SDC-2 axis's role in EMT, the mechanistic insights are somewhat superficial. The focus is heavily on EMT marker changes without adequately addressing the epithelial cells’ functional outcome, specifically related to cystitis. More extensive exploration on downstream signaling pathways related to HPSE and SDC-2 in EMT, perhaps involving other known mediators, could strengthen the paper’s mechanistic foundation and relevance for IC/BPS treatment.

• Along the same lines, ambiguity in experimental setup and the methodology lacks clarity in certain areas. For example, in Results section 3.1, paragraph two, you indicate an intention to study the relationship between EMT and cellular function in vitro, using an experiment based on LPS treatment (presumably involving LPS injection in mice, though this is not explicitly stated). However, it is unclear how this approach directly addresses the research question. Later, a conclusion is drawn that LPS induces EMT, but this does not fully address the specific question of epithelial cell function, leaving the connection insufficiently demonstrated.

• The figures predominantly show a reduction in epithelial marker expression, accompanied by an increase in mesenchymal markers, which suggests EMT induction. However, additional clarification would strengthen this interpretation.

• Much of the data relies on pharmacological inhibition of HPSE via OGT2115, but no complementary genetic knockdown model (e.g., siRNA against HPSE).

• Specific details are missing, such as the number of mice used for each experiment and the control conditions, including necessary controls for siRNA (e.g., Si-scramble control for SDC-2 knockdown). The absence of these details weakens the methodological rigor

• Additionally, there is a lack of precision in terms of the number of mice used in each experiment. On page 5, Section 2.12, the statistical methods section mentions standard deviations and ANOVA tests but lacks detailed information on sample sizes for each experiment. This makes it challenging to assess the robustness of the findings. Please address these issues to enhance the rigor of your research and provide more details on group sizes, randomization, also blinding methods, particularly regarding pain and urodynamic assessments

• Several figures, including 2A-B, lack clarity and are sometimes unreadable, which diminishes the impact and interpretability of the results. High-resolution images should be provided to ensure that all figure details are visible. Furthermore, the contrast and clarity of fluorescence or histological images should be improved to enhance figure readability.

6. PLOS authors have the option to publish the peer review history of their article (what does this mean?). If published, this will include your full peer review and any attached files.

Reviewer #1: No

Reviewer #2: **Yes: **LEBSIR Nadjet

---

## [Author Response · Author response to Decision Letter 1]

15 Dec 2024

Dear Editors and Reviewers:

Thank you for your letter and for the reviewers’ comments concerning our manuscript entitled “Inhibition of HPSE/SDC-2 axis-induced epithelial-mesenchymal transition for treating IC/BPS” (ID: PONE-D-24-45975). Those comments are all valuable and helpful for revising and improving our paper, as well as the important guiding significance to our research. We have studied the comments carefully and have made corrections, which we hope to meet with approval.

The revisions in the manuscript are in red fonts, and responses to the reviewers' comments are in blue fonts.

Reviewer #1: In their study, Chen et al. describe the involvement of the HPSE/SDC-2 axis in the development of epithelial-to-mesenchymal transition (EMT) in interstitial cystitis/bladder pain syndrome (IC/BPS) and propose a new therapeutic target for this pathology. In vivo and in vitro experiments were conducted to test the small molecule OGT2115, which was confirmed to inhibit the EMT process. The in vitro studies indicate the potential involvement of HPSE and SDC-2 in the development of EMT and suggest that their inhibition may facilitate the restoration of a favorable phenotype. The authors conclude that OGT2215, an HPSE inhibitor, may represent a promising therapeutic molecule for IC/PBS.

I have a few comments/suggestions to improve and strengthen the study:

• The authors used OGT2115, a HPSE-1 inhibitor, to unveil HPSE-1 role on EMT. The choice of 5 µM as an optimal drug concentration was based solely on EMT phenotypic changes (decrease of E-cadherin and increase of Vimentin at protein level). However, Western blots of Fig 4A-B do not show a strong inhibition of HPSE-1 expression at this concentration. What is the IC50 of OGT2115 on HPSE-1 expression?

Response to comment: Thanks for the comments. OGT2115 (HY-100898, MCE) is a reliable HPSE-1 inhibitor, and its product description indicates that it has an IC50 of 0.4 μM for HPSE-1 inhibition and an IC50 of 1μM for antiangiogenicity. Our reading of the literature has revealed that most of the intervening concentrations of OGT2115 used at in vitro levels are 1 μM ( PMID: 32608014, PMID: 31327483). However, in our pre-experiment results, we found that OGT2115 did not significantly improve the EMT occurring in Sv-huc-1 when intervened at 1 μM, which may be due to the difference in cell lines, different cell culture conditions, etc. Therefore, we set up 0,1.25, 2.5, 5, 10 and 20μM concentration gradients to explore the optimal action concentration of OGT2115 for EMT improvement. According to the purpose of our study, i.e., the ameliorating effect of OGT2115 on EMT, we chose an intervention concentration of 5μM and verified the inhibitory effect of OGT2115 on the HPSE/SDC-2 axis at this concentration in Figure 4C. The reviewer's suggestion was very inspiring, and we will plot IC50 curves directly against the drug's target of action in future studies.

• It is important to perform cytotoxicity assays of OGT2115 to confirm the non-cytotoxic effect of the drug.

Response to comment: Thanks for the comments. We determined the cytotoxicity of OGT2115 using the CCK-8 assay. The results showed that for Sv-huc-1, the IC50 of OGT2115 was 36.26 μM. 5 μM is a safe intervention concentration that does not significantly inhibit cell viability.

• Title and conclusion of the 3.4 paragraph are not very clear. Is there an improvement or an inhibition of the EMT process under OGT2215 treatment? Clarify your message.

Response to comment: Thanks for the comments, and we have described this result more clearly.

• The authors focused on the HPSE/SDC-2 axis implication by modulating HPSE (through OGT2115) but to strengthen and confirm their observations, the use of SDC-2 inhibitors could be interesting. Similarly, the authors performed RNAi experiments with only SDC-2 RNAi. It would be interesting to genetically modulate HPSE expression (by siRNA), deplete both SDC-2/HPSE and finally rescue their expression, to confirm that both proteins are essential.

Response to comment: Thanks for the comments. We searched for SDC-2 inhibitors in the preliminary stage but did not find any product in sale. We refined the overexpression of HPSE and the related response experiments according to the reviewers' suggestions. The experimental results are elaborated as follows:

“We overexpressed HPSE and validated it using WB (Figure S1A). Overexpression of HPSE promoted EMT progression, epithelial dysfunction, and increased fluorescence leakage occurred (Figure S1B-D). We knocked down SDC-2 based on overexpression of HPSE (Figure S1E). The EMT-promoting effect exerted by overexpression of HPSE was counteracted by SDC-2 knockdown (Figure S1E-F). The epithelial function was restored and fluorescence leakage was reduced after the SDC-2 knockdown (Figure S1G). Combined with the previous results, it can be concluded that HPSE binding to SDC-2 caused downstream EMT development and epithelial dysfunction, with SDC-2 playing a key role.”

• In their study, the authors analyzed GSE57560 dataset and described an upregulation for SDC-2 but also for SDC-1 and -4. However, they did not go further with these receptors because there are not significant changes in mice. Can the authors discuss more on that aspect? What is the difference between human and mice for SDC expression?

Response to comment: Thanks for the comments. Mice are able to mimic human pathophysio -logical processes to some extent, but the distribution and function of some proteins such as the SDC family in the bladder are not identical. We analyzed human bladder tissue high-throughput sequencing data and found that the HPSE/SDC axis may play a role in the disease process. From the experimental results, human bladder tissues SDC1, SDC4 were also significantly elevated and had higher expression during the disease process of IC. We found low and no significant difference in SDC1, and SDC4 expression in the mouse model.

We observed the correlation between EMT and urination in mice from a mouse model. The role of EMT on phenotype was verified from the perspective of altered urinary habits. This part of the experimental results of altered urination habits provided by the mouse model is not available in human cell lines, so we will choose SDC-2, which is expressed differently and significantly in mice, for subsequent studies. We look forward to observing the role of the HPSE/SDC-2 axis in the altered urinary behavior of mice in inflammatory states by treating the mice again accordingly, provided that the in vitro levels are verified exactly.

• As they use human cell lines, it could be interesting to perform some experiments in vitro on SDC-1 and -4. Do they interact with HPSE? Are they involved in EMT process? Can they partially replace SDC-2 when SDC-2 is knockdown?

Response to comment: Thanks for the comments. As stated above we chose SDC-2 for subsequent experiments, and EMT was significantly suppressed and restored to near-normal levels when we treated SDC-2 with knockdown (Figure 5D-F). This result suggests that SDC-2 is a key intermediate protein for HPSE to play a role in promoting EMT and is consistent with the results of the mouse model. We strongly agree with the reviewer's suggestion to refine the alternative roles of other proteins of the SDC family in the SDC-2 knockdown state, and we will further refine this part of the relevant content in subsequent studies.

• Finally, they only used phenotypic markers to describe the EMT process (E-cadherin as epithelial marker and Vimentin for mesenchymal marker), but they did not control cytokine expression, already described in the literature to be involved in this process (e.g. TGF-β). In the discussion part, the authors suggest that HPSE can activate TGF-β (without citation). Regarding these informations, it could be interesting to add a TGF-β inhibitor to confirm the EMT induction is TFG-β -dependent.

Response to comment: Thanks for the comments. We subsequently refined the downstream mechanistic studies that SDC-2 is able to activate the TGF-β pathway by binding to SMAD7 and reducing TGFβR1 ubiquitination degradation. This part has been added to the manuscript:

“The TGF-β pathway is an important pathway for promoting EMT and a downstream signaling pathway for HPSE. We found that it was significantly activated in the LPS intervention group and inhibited after the knockdown of SDC-2 (Figure 6D). Then we exogenously added TGF-β1 as a pathway activator, and we found that the activation of the pathway by TGF-β1 disappeared after SDC-2 knockdown (Figure 6E). The activating effect of TGF-β1 was further enhanced after overexpression of SDC-2 (Figure 6F). This result suggests that the activating effect of SDC-2 on the TGF-β pathway is not through the promotion of TGF-β1 secretion, but may exert its biological function through its receptor. Ubiquitination is one of the pathways that promote protein degradation, and we examined the level of ubiquitination of TGF-βR1 under knockdown and overexpression of SDC-2. The results showed that the ubiquitination level of TGF-βR1 increased after the knockdown of SDC-2, and the ubiquitination level of TGF-βR1 decreased after the overexpression of SDC-2 (Figure 6G-H). Taken together, we can conclude that SDC-2 can inhibit TGF-β1 ubiquitination degradation thereby activating the TGF-β pathway.”

Reviewer #2: The manuscript investigates the role of HPSE (heparanase) and SDC-2 in epithelial-mesenchymal transition (EMT) within the context of interstitial cystitis/bladder pain syndrome (IC/BPS), proposing that the inhibition of the HPSE/SDC-2 axis could mitigate EMT-related epithelial dysfunction. While the study presents intriguing data and utilizes multiple methodologies (in vivo, in vitro), there are critical areas where improvements are necessary :

in the introduction section :

• It might be helpful to add specific epidemiological data on IC/BPS, like prevalence or challenges in diagnosis. Including a bit more detail on current treatments and their limitations could help to show why targeting EMT and HPSE/SDC-2 might offer something new in terms of therapy.

Response to comment: Thanks for the comments, and we have added this section.

• Adding a few lines on the functions of HPSE and SDC-2, especially their roles in extracellular matrix remodeling and EMT, might help readers unfamiliar with these molecules to understand why they matter in IC/BPS

Response to comment: Thanks for the comments, and we have added this section.

• The hypothesis is implicit rather than clearly stated. While the introduction discusses the potential involvement of HPSE and SDC-2 in EMT in IC/BPS, it lacks an explicit hypothesis that clearly directs the study's objectives. e.g., “We hypothesize that the HPSE/SDC-2 axis promotes EMT in IC/BPS and that its inhibition can ameliorate EMT-associated epithelial dysfunction.”.

Response to comment: Thanks for the comments. We have formulated hypotheses in the original article to explicitly guide the research objectives.

• You should add a few more references specifically for EMT in bladder disease could strengthen your argument. For exemple, citing related literature on EMT in non-cancerous inflammatory conditions would more enhance the study’s novelty.

Response to comment: Thanks for the comments. We have added EMT-related references.

Methodology and results

• The Results section should be restructured for clarity and logical flow. Each sub-section should begin with a brief statement of specific objectives, followed by a concise summary of the methods used.

Response to comment: Thanks for the comments, and we have made amendments.

• Also, there is sometimes disorganized presentation of findings, for exemple, the abrupt presentation of EMT in bladder tissues, before first confirming cystitis validation, disrupts the logical progression of the results. This sequence should ideally be reversed to establish the model before discussing EMT induction.

Response to comment: Thanks for the comments. We placed the phenotypic validation at the front of the narrative to refine the logic of the study.

• Although the manuscript discusses the HPSE/SDC-2 axis's role in EMT, the mechanistic insights are somewhat superficial. The focus is heavily on EMT marker changes without adequately addressing the epithelial cells’ functional outcome, specifically related to cystitis. More extensive exploration on downstream signaling pathways related to HPSE and SDC-2 in EMT, perhaps involving other known mediators, could strengthen the paper’s mechanistic foundation and relevance for IC/BPS treatment.

Response to comment: Thanks for the comments.We are very sorry that we may have been misleading in the labeling of some experimental sections. Our functional verification of the epithelial cells was mainly done by the epithelial permeability assay, which is the measurement of their anti-leakage function to fluorescent reagents 24h after cell passaging. High fluorescence intensity represents that more fluorescence leakage occurs and epithelial dysfunction.

We explored the downstream pathways and describe them below in the manuscript:

“The TGF-β pathway is an important pathway for promoting EMT and a downstream signaling pathway for HPSE. We found that it was significantly activated in the LPS intervention group and inhibited after the knockdown of SDC-2 (Figure 6D). Then we exogenously added TGF-β1 as a pathway activator, and we found that the activation of the pathway by TGF-β1 disappeared after SDC-2 knockdown (Figure 6E). The activating effect of TGF-β1 was further enhanced after overexpression of SDC-2 (Figure 6F). This result suggests that the activating effect of SDC-2 on the TGF-β pathway is not through the promotion of TGF-β1 secretion, but may exert its biological function through its receptor. Ubiquitination is one of the pathways that promote protein degradation, and we examined the level of ubiquitination of TGF-βR1 under knockdown and overexpression of SDC-2. The results showed that the ubiquitination level of TGF-βR1 increased after the knockdown of SDC-2, and the ubiquitination level of TGF-βR1 decreased after the overexpression of SDC-2 (Figure 6G-H). Taken together, we can conclude that SDC-2 can inhibit TGF-β1 ubiquitination degradation thereby activating the TGF-β pathway.”

• Along the same lines, ambiguity in experimental setup and the methodology lacks clarity in certain areas. For example, in Results section 3.1, paragraph two, you indicate an intention to study the relationship between EMT and cellular function in vitro, using an experiment based on LPS treatment (presumably involving LPS injection in mice, though this is not explicitly stated). However, it is unclear how this approach directly addresses the research question. Later, a conclusion is drawn that LPS induces EMT, but this does not fully address the specific question of epithelial cell function, leaving the connection insufficiently demonstrated.

Response to comment: Thanks for the comments. We apologize for the misunderstanding caused by the unclear narrative. The second paragraph of Result 3.1 was validated using human-derived Sv-huc-1 at the in vitro level. We plotted the correlation curve using the WB relative protein expression of the EMT indicator as the horizontal coordinate and the fluorescence intensity at which leakage occurs as the vertical coordinate.

• The figures predominantly show a reduction in epithelial marker expression, accompanied by an increase in mesenchymal markers, which suggests EMT induction. However, additional clarification would strengthen this interpretation.

Response to comment: Thanks for the comments. The current confirmation of EMT in non-tumorigenic diseases mainly rests on functional validation, so we first validated the effect of EMT on urination in mice and the effect of EMT on the barrier function of epithelial cells in Figure 1. In the subsequent validation process we also validated the progression of EMT by altered urination in mice (Figure 2D-I), and epithelial cell anti-leakage function (Figure 4I, Figure 5I).

• Much of the data relies on pharmacological inhibition of HPSE via OGT2115, but no complementary genetic knockdown model (e.g., siRNA against H

---

## [Decision Letter · Decision Letter 1]

14 Jan 2025

PONE-D-24-45975R1Inhibition of HPSE/SDC-2 axis-induced epithelial-mesenchymal transition for treating IC/BPSPLOS ONE

Dear Dr. He,

Thank you for submitting your manuscript to PLOS ONE. After careful consideration, we feel that it has merit but does not fully meet PLOS ONE’s publication criteria as it currently stands. Therefore, we invite you to submit a revised version of the manuscript that addresses the points raised during the review process.

You should address the questions raised by the reviewers, including improvement of the introduction section with epidemiological data, improvement of the results, specifying that the study concerns HPSE-1, explaining abbreviations, distinguishing between mice and human data, OGT2115 cytotoxicity, etc.

We look forward to receiving your revised manuscript.

Kind regards,

Boyan Grigorov

Academic Editor

PLOS ONE

Reviewers' comments:

Reviewer's Responses to Questions

**Comments to the Author**

1. If the authors have adequately addressed your comments raised in a previous round of review and you feel that this manuscript is now acceptable for publication, you may indicate that here to bypass the “Comments to the Author” section, enter your conflict of interest statement in the “Confidential to Editor” section, and submit your "Accept" recommendation.

Reviewer #1: (No Response)

2. Is the manuscript technically sound, and do the data support the conclusions?

Reviewer #1: Partly

3. Has the statistical analysis been performed appropriately and rigorously? 

Reviewer #1: I Don't Know

4. Have the authors made all data underlying the findings in their manuscript fully available?

Reviewer #1: Yes

5. Is the manuscript presented in an intelligible fashion and written in standard English?

Reviewer #1: Yes

6. Review Comments to the Author

Reviewer #1: Thank you for considering my previous comments. However, there are still several issues that need to be addressed:

Introduction Section:

• You added some epidemiological data as requested by Reviewer #2, but it would be helpful to provide more specific details, particularly regarding prevalence data (e.g., breakdown by gender, age, and geographical region).

• To ensure clarity for unfamiliar readers, please specify that the study focuses on HPSE-1 (the enzymatically active form), rather than HPSE-2.

Results Section:

• Abbreviations are missing in some areas. It would be beneficial to define them in the results section for easier readability (e.g., CYP mice are described in the Materials and Methods section but could be redefined at the beginning of the paragraph). Additionally, abbreviations in the figures, such as NC, CON, etc., should be included.

• As per Reviewer #2's request, it would be helpful to include a brief introduction at the beginning of each subsection, outlining the specific objectives of the experiments and the methods used (e.g., "To answer this question, we employed the following methods...").

Section 3.1:

• The title of this section is misleading because it discusses results from mice, not just in vitro experiments.

• It would be helpful to briefly explain the use of LPS for in vitro studies. Although this information is in the Materials and Methods section, it would be clearer if restated in the Results section.

• The conclusion in this section is somewhat confusing. While the results pertain to in vitro studies, the conclusion seems to mix in vivo and in vitro findings. A summary of both in vivo and in vitro results would make this clearer before the conclusion.

Section 3.2:

• The conclusion specifies the role of OGT2115 in improving urinary symptoms. It would be beneficial to include the name of the inhibitor in the section title for clarity.

• It should be clearly stated when analyses were performed on human samples compared to mice. For example, is the GSE57560 dataset from human or mouse samples?

• For Figures E and F, it would be helpful to add more detailed legends to aid in the interpretation of the data.

Section 3.3:

• Figure 3A demonstrates a downregulation of HPSE under OGT2115 treatment, not inhibition. Please use the correct terminology.

• It may be useful to compare the condition with OGT2115 to control samples to determine whether there is a full rescue of the epithelial phenotype with the treatment (Figures 3B-C and 3E-G).

Section 3.4:

• OGT2115 is used to inhibit HPSE in vitro. Cytotoxicity assays should be included in this section (or in supplementary data), and the controls need to be clearly specified. Was OGT2115 resuspended in PBS or DMSO? How were the CCK-8 assay results normalized? Is cytotoxicity caused by DMSO or by OGT2115 itself? Furthermore, in the authors' responses to previous requests, their graph did not display the data for 50% cell mortality. Did they extrapolate this data (e.g., 36.26 µM determined as the dose for 50% cell death)? Greater rigor is needed in this experiment.

• In this section, HPSE activity is modulated through pharmacological inhibition, but to strengthen the results, HPSE knockdown experiments should complement these findings (as requested by both Reviewer #1 and Reviewer #2). HPSE knockdown has been proposed as a method to confirm OGT2115's effects and exclude potential off-target effects. However, the HPSE overexpression experiments already conducted are useful and strengthen the results.

Section 3.6:

• Thank you for incorporating our comments to investigate a potential mechanism by exploring the TGF-β pathway.

• The first paragraph could be better integrated into the previous one discussing the role of SDC-2 and its interaction with HPSE for more coherence.

• For readers who may be unfamiliar with the TGF-β pathway, it would be helpful to briefly describe it and explain the rationale behind studying Smad2.

• The sentence, "This result suggests that the activating effect of SDC-2 on the TGF-β pathway is not through the promotion of TGF-β1 secretion, but may exert its biological function through its receptor," would benefit from further clarification. Please explain the underlying hypothesis more explicitly.

7. PLOS authors have the option to publish the peer review history of their article (what does this mean?). If published, this will include your full peer review and any attached files.

Reviewer #1: No

---

## [Author Response · Author response to Decision Letter 2]

10 Feb 2025

Dear Editors and Reviewers:

Thank you for your letter and for the reviewers’ comments concerning our manuscript entitled “Inhibition of HPSE/SDC-2 axis-induced epithelial-mesenchymal transition for treating IC/BPS” (ID: PONE-D-24-45975). Those comments are all valuable and helpful for revising and improving our paper, as well as the important guiding significance to our research. We have studied the comments carefully and have made corrections, which we hope to meet with approval.

The revisions in the manuscript are in red fonts, and responses to the reviewers' comments are in blue fonts.

Reviewer #1: Thank you for considering my previous comments. However, there are still several issues that need to be addressed:

Introduction Section:

• You added some epidemiological data as requested by Reviewer #2, but it would be helpful to provide more specific details, particularly regarding prevalence data (e.g., breakdown by gender, age, and geographical region).

Response to comment: Thanks for the comments. We have added epidemiological data as follows:

“The prevalence of IC/BPS ranges from approximately 2.7% to 6.5%, with women being affected at a rate ten times higher than men. The global prevalence of IC/BPS is approximately 300 per 100,000 women, with the prevalence among men being about 10%-20% of that in women. When considering only the presence of symptoms suggestive of IC/BPS, the incidence rates for both men and women may increase by more than 10 times.”

• To ensure clarity for unfamiliar readers, please specify that the study focuses on HPSE-1 (the enzymatically active form), rather than HPSE-2.

Response to comment: Thanks for the comments. We have specified that our study focused on HPSE-1 as follows:

“There are two forms of heparinase: heparinase-1 (HPSE) and heparinase-2, with heparinase-1 exerting the primary biological functions, which our study focused on.”

Results Section:

• Abbreviations are missing in some areas. It would be beneficial to define them in the results section for easier readability (e.g., CYP mice are described in the Materials and Methods section but could be redefined at the beginning of the paragraph). Additionally, abbreviations in the figures, such as NC, CON, etc., should be included.

Response to comment: Thanks for the comments. We have added this section.

• As per Reviewer #2's request, it would be helpful to include a brief introduction at the beginning of each subsection, outlining the specific objectives of the experiments and the methods used (e.g., "To answer this question, we employed the following methods...").

Response to comment: Thanks for the comments. We have added this section.

Section 3.1:

• The title of this section is misleading because it discusses results from mice, not just in vitro experiments.

Response to comment: Thanks for the comments. We have corrected the following in the original text:

“EMT correlates with urination frequency in mice and barrier function in urothelial cells.”

• It would be helpful to briefly explain the use of LPS for in vitro studies. Although this information is in the Materials and Methods section, it would be clearer if restated in the Results section.

Response to comment: Thanks for the comments. We have added this section as follows:

“We then constructed an urothelial EMT model in vitro by continuously stimulating Sv-huc-1 with LPS for 24 hours to further investigate the relationship between EMT and urothelial cell function.”

• The conclusion in this section is somewhat confusing. While the results pertain to in vitro studies, the conclusion seems to mix in vivo and in vitro findings. A summary of both in vivo and in vitro results would make this clearer before the conclusion.

Response to comment: Thanks for the comments. We have added this section as follows:

“This result demonstrates a correlation between the occurrence of EMT and urothelial cell barrier function, suggesting that the progression of EMT disrupts the epithelial cell barrier function. Taken together, these experimental findings indicate that EMT is associated with urination frequency in mice and barrier function in urothelial cells.”

Section 3.2:

• The conclusion specifies the role of OGT2115 in improving urinary symptoms. It would be beneficial to include the name of the inhibitor in the section title for clarity.

Response to comment: Thanks for the comments. We have added this section.

• It should be clearly stated when analyses were performed on human samples compared to mice. For example, is the GSE57560 dataset from human or mouse samples?

Response to comment: Thanks for the comments. We have added this section as follows:

“The interstitial cystitis dataset GSE57560 was downloaded from the GEO public database, containing 12 interstitial cystitis and 4 control bladder biopsy tissues from humans.”

• For Figures E and F, it would be helpful to add more detailed legends to aid in the interpretation of the data.

Response to comment: Thanks for the comments. We have added this section as follows:

“E: Fixed volumes (5, 10, 20, 50, 100 μl) of mouse urine were dropped onto filter paper for experimental use. ImageJ was used to calculate the area, and an area-volume standard curve was plotted, resulting in the formula: Volume = Area * 12.674. F: Based on the area of mouse urine spots, quantitative analysis was conducted on the frequency of urination, average urine volume, and total urine volume in mice (n=5).”

Section 3.3:

• Figure 3A demonstrates a downregulation of HPSE under OGT2115 treatment, not inhibition. Please use the correct terminology.

Response to comment: Thanks for the comments. We have corrected the following in the original text:

“At the in vivo level by WB assay, OGT2115 was shown to down-regulate the expression of HPSE and SDC-2 (Fig 3A).”

• It may be useful to compare the condition with OGT2115 to control samples to determine whether there is a full rescue of the epithelial phenotype with the treatment (Figures 3B-C and 3E-G).

Response to comment: Thanks for the comments. We have added this section.

Section 3.4:

• OGT2115 is used to inhibit HPSE in vitro. Cytotoxicity assays should be included in this section (or in supplementary data), and the controls need to be clearly specified. Was OGT2115 resuspended in PBS or DMSO? How were the CCK-8 assay results normalized? Is cytotoxicity caused by DMSO or by OGT2115 itself? Furthermore, in the authors' responses to previous requests, their graph did not display the data for 50% cell mortality. Did they extrapolate this data (e.g., 36.26 µM determined as the dose for 50% cell death)? Greater rigor is needed in this experiment.

Response to comment: Thanks for the comments. We did not validate the cytotoxicity of OGT2115 at higher concentrations since the highest concentration of OGT2115 we used was 20 μM. We plotted fitting curves and extrapolated IC50 values based on the available data. We have added this section as follows:

“Initially, we used the CCK8 method to measure the cytotoxicity of OGT2115, with the control group receiving DMSO as the solvent. After CCK8 incubation, we measured the OD values, subtracted the blank well value (only culture medium added) from each well, and divided by the control group to obtain the percentage of live cells. After importing the data into Prism, we plotted a fitted curve using the logarithm of concentration as the x-axis and cell survival rate as the y-axis. From this, we calculated an IC50 value of 36.26 μM (Figure 4A).”

• In this section, HPSE activity is modulated through pharmacological inhibition, but to strengthen the results, HPSE knockdown experiments should complement these findings (as requested by both Reviewer #1 and Reviewer #2). HPSE knockdown has been proposed as a method to confirm OGT2115's effects and exclude potential off-target effects. However, the HPSE overexpression experiments already conducted are useful and strengthen the results.

Response to comment: We thank the reviewers for their acknowledgement of our experimental protocol. Since an inhibitor of HPSE was used in the experiment, we considered overexpression of HPSE in the subsequent validation to better illustrate the reliability of the conclusions.

Section 3.6:

• Thank you for incorporating our comments to investigate a potential mechanism by exploring the TGF-β pathway.

• The first paragraph could be better integrated into the previous one discussing the role of SDC-2 and its interaction with HPSE for more coherence.

Response to comment: Thanks for the comments. We have added this section as follows:

“The above experiments demonstrate that SDC-2 is a downstream protein that binds to HPSE, but the specific binding site remains unclear. Therefore, based on the HS chain region of SDC-2, we designed truncations (Fig 6A).”

• For readers who may be unfamiliar with the TGF-β pathway, it would be helpful to briefly describe it and explain the rationale behind studying Smad2.

Response to comment: Thanks for the comments. We have added this section as follows:

“Transforming growth factor-β (TGF-β) is a profound factor that promotes fibrosis and EMT, and it is also a downstream signaling pathway of HPSE. When TGF-β1 binds to its receptor, TGFβR1, it can phosphorylate the downstream protein SMAD2. Phosphorylated SMAD2 can bind to SMAD4 and enter the nucleus, activating the signaling pathway through the canonical pathway.”

• The sentence, "This result suggests that the activating effect of SDC-2 on the TGF-β pathway is not through the promotion of TGF-β1 secretion, but may exert its biological function through its receptor," would benefit from further clarification. Please explain the underlying hypothesis more explicitly.

Response to comment: Thanks for the comments. We have added this section as follows:

“The above results show that even with exogenous addition of TGF-β1, the TGFβ signaling pathway cannot be significantly activated after knocking down SDC-2, while the activation effect of TGF-β1 is significantly enhanced after overexpressing SDC-2.”

---

## [Decision Letter · Decision Letter 2]

4 Mar 2025

PONE-D-24-45975R2Inhibition of HPSE/SDC-2 axis-induced epithelial-mesenchymal transition for treating IC/BPSPLOS ONE

Dear Dr. He,

Thank you for submitting your manuscript to PLOS ONE. After careful consideration, we feel that it has merit but does not fully meet PLOS ONE’s publication criteria as it currently stands. Therefore, we invite you to submit a revised version of the manuscript that addresses the points raised during the review process.

Both reviewers noted that hepanase was not written correctly within the text. Be careful to replace all wrong words. Also improve the results section as suggested.

We look forward to receiving your revised manuscript.

Kind regards,

Boyan Grigorov

Academic Editor

PLOS ONE

Journal Requirements:

Reviewers' comments:

Reviewer's Responses to Questions

**Comments to the Author**

1. If the authors have adequately addressed your comments raised in a previous round of review and you feel that this manuscript is now acceptable for publication, you may indicate that here to bypass the “Comments to the Author” section, enter your conflict of interest statement in the “Confidential to Editor” section, and submit your "Accept" recommendation.

Reviewer #1: (No Response)

Reviewer #3: (No Response)

2. Is the manuscript technically sound, and do the data support the conclusions?

Reviewer #1: (No Response)

Reviewer #3: Partly

3. Has the statistical analysis been performed appropriately and rigorously? 

Reviewer #1: (No Response)

Reviewer #3: N/A

4. Have the authors made all data underlying the findings in their manuscript fully available?

Reviewer #1: (No Response)

Reviewer #3: Yes

5. Is the manuscript presented in an intelligible fashion and written in standard English?

Reviewer #1: (No Response)

Reviewer #3: No

6. Review Comments to the Author

Reviewer #1: Thank you for taking my previous comments into consideration. The reading and comprehension are improving, but there are still a few mistakes or misunderstandings that need to be addressed.

- Please be careful: in the introduction section, you mention "heparinase" instead of "heparanase." This needs to be corrected.

- In section 3.1, I would recommend explaining the CYP group further (What does cyclophosphamide do in mice? Which phenotype does it induce?), similar to the explanation already provided for LPS treatment on Sv-huc-1 cells.

- Be cautious, as some information is repeated within the same paragraph (e.g., "We added LPS stimulation to construct a cellular EMT model" and "We constructed an EMT model of urothelial cells by stimulating them continuously with LPS for 24 hours in the culture medium"). Avoid such repetitions to improve readability.

Reviewer #3: The study has its merits but more efforts need to be made for the presentation of the manuscript. In the abstract it is not clear what CYP-induced mean? What does the abbreviation mean? Cyclophosphamide? Cyclophilin?

Be careful when writing heparanase, since I saw "heparinase" with "i" many times in the text. This is not the same enzyme. Replace where needed.

An effort should be made to present the results in a more comprehensive manner. For example, look at the first paragraph of the "Results" section: the authors should explain what is the objective/rationale of the experiment, why they do it (what is the expected outcome) and then to explain how (western blot and voiding spots).

In the "Results" section, the title of the following paragraph is not clear: “Inhibition of HPSE ameliorates cystogenesis in CYP-induced cystitis mice EMT”. Also, explain why you check E-cadherin, N-cadherin, vimentin…

7. PLOS authors have the option to publish the peer review history of their article (what does this mean?). If published, this will include your full peer review and any attached files.

Reviewer #1: No

Reviewer #3: No

---

## [Author Response · Author response to Decision Letter 3]

5 Mar 2025

Dear Editors and Reviewers:

Thank you for your letter and for the reviewers’ comments concerning our manuscript entitled “Inhibition of HPSE/SDC-2 axis-induced epithelial-mesenchymal transition for treating IC/BPS” (ID: PONE-D-24-45975). Those comments are all valuable and helpful for revising and improving our paper, as well as the important guiding significance to our research. We have studied the comments carefully and have made corrections, which we hope to meet with approval.

The revisions in the manuscript are in red fonts, and responses to the reviewers' comments are in blue fonts.

Reviewer #1: Thank you for taking my previous comments into consideration. The reading and comprehension are improving, but there are still a few mistakes or misunderstandings that need to be addressed.

- Please be careful: in the introduction section, you mention "heparinase" instead of "heparanase." This needs to be corrected.

Response to comment: Thanks for the comments. We have corrected this error in the introduction section.

- In section 3.1, I would recommend explaining the CYP group further (What does cyclophosphamide do in mice? Which phenotype does it induce?), similar to the explanation already provided for LPS treatment on Sv-huc-1 cells.

Response to comment: Thanks for the comments. We have added the biological functions performed by CYP in the original text, which are described as follows:

“Following intraperitoneal injection, cyclophosphamide (CYP) is metabolized in the kidney into acrolein, a toxic compound to urothelial cells. This process induces urothelial injury and inflammation, ultimately triggering EMT.”

- Be cautious, as some information is repeated within the same paragraph (e.g., "We added LPS stimulation to construct a cellular EMT model" and "We constructed an EMT model of urothelial cells by stimulating them continuously with LPS for 24 hours in the culture medium"). Avoid such repetitions to improve readability.

Response to comment: Thanks for the comments. We have removed the repetitive language.

Reviewer #3: The study has its merits but more efforts need to be made for the presentation of the manuscript. In the abstract it is not clear what CYP-induced mean? What does the abbreviation mean? Cyclophosphamide? Cyclophilin?

Response to comment: Thanks for the comments. We have included the full name of cyclophosphamide (CYP) in the abstract section.

“Both the increased urination frequency observed in mice with acute cystitis induced by cyclophosphamide (CYP) and the disruption of the anti-leakage barrier in urothelial cells induced by LPS are associated with the occurrence of EMT.”

Be careful when writing heparanase, since I saw "heparinase" with "i" many times in the text. This is not the same enzyme. Replace where needed.

Response to comment: Thanks for the comments. We have rectified this mistake, and we thank the reviewer for pointing it out.

An effort should be made to present the results in a more comprehensive manner. For example, look at the first paragraph of the "Results" section: the authors should explain what is the objective/rationale of the experiment, why they do it (what is the expected outcome) and then to explain how (western blot and voiding spots).

Response to comment: Thanks for the comments. We have added an elaboration on the results in the first paragraph of the results section, stated as follows:

“To investigate the correlation between the frequency of urination in mice and epithelial-mesenchymal transition (EMT), we quantified the number of urinations by counting the urine spots on filter paper in the voiding spots experiment.”

“The epithelial permeability assay was employed to evaluate the barrier function of urothelial cells, where the fluorescent intensity of leakage is inversely proportional to the barrier function.”

In the "Results" section, the title of the following paragraph is not clear: “Inhibition of HPSE ameliorates cystogenesis in CYP-induced cystitis mice EMT”. Also, explain why you check E-cadherin, N-cadherin, vimentin…

Response to comment: Thanks for the comments. E-cadherin, N-cadherin, and Vimentin are three EMT-related proteins that are widely used and validated in EMT-related studies. During our literature review, we found that previous studies on IC/BPS and EMT had adopted these three proteins, which were verified to be expressed in urothelial cells (PMID: 34368980, PMID: 34984178). Therefore, we conducted our research using E-cadherin, N-cadherin, and Vimentin.

---

## [Editor Report · Decision Letter 3]

11 Mar 2025

Inhibition of HPSE/SDC-2 axis-induced epithelial-mesenchymal transition for treating IC/BPS

PONE-D-24-45975R3

Dear Dr. He,

We’re pleased to inform you that your manuscript has been judged scientifically suitable for publication and will be formally accepted for publication once it meets all outstanding technical requirements.

Kind regards,

Boyan Grigorov

Academic Editor

PLOS ONE
---

## [Editor Report · Acceptance letter]

PONE-D-24-45975R3

PLOS ONE

Dear Dr. He,

I'm pleased to inform you that your manuscript has been deemed suitable for publication in PLOS ONE. Congratulations! Your manuscript is now being handed over to our production team.

Kind regards,

on behalf of

Dr. Boyan Grigorov

Academic Editor

PLOS ONE